# Effects of Protein Supplementation Combined with Resistance Exercise Training on Walking Speed Recovery in Older Adults with Knee Osteoarthritis and Sarcopenia

**DOI:** 10.3390/nu15071552

**Published:** 2023-03-23

**Authors:** Chun-De Liao, Shih-Wei Huang, Hung-Chou Chen, Yu-Yun Huang, Tsan-Hon Liou, Che-Li Lin

**Affiliations:** 1International Ph.D. Program in Gerontology and Long-Term Care, College of Nursing, Taipei Medical University, Taipei 110301, Taiwan; 08415@s.tmu.edu.tw; 2Department of Physical Medicine and Rehabilitation, Shuang Ho Hospital, Taipei Medical University, New Taipei City 235041, Taiwan; 13001@s.tmu.edu.tw (S.-W.H.); 10462@s.tmu.edu.tw (H.-C.C.); peter_liou@s.tmu.edu.tw (T.-H.L.); 3Department of Physical Medicine and Rehabilitation, School of Medicine, College of Medicine, Taipei Medical University, Taipei 110301, Taiwan; 4Department of Pediatrics, New York University Langone Medical Center, New York City, NY 10016, USA; huang-yu-yun@hotmail.com; 5Department of Orthopedic Surgery, Shuang Ho Hospital, Taipei Medical University, New Taipei City 23561, Taiwan; 6Department of Orthopedics, School of Medicine, College of Medicine, Taipei Medical University, Taipei 11031, Taiwan

**Keywords:** sarcopenia, osteoarthritis, protein supplement, exercise, mobility

## Abstract

Knee osteoarthritis (KOA) is closely associated with sarcopenia, sharing the common characteristics of muscle weakness and low physical performance. Resistance exercise training (RET), protein supplementation (PS), and PS+RET have promise as treatments for both sarcopenia and KOA. However, whether PS+RET exerts any effect on time to recovery to normal walking speed (WS) in older adults with sarcopenia and KOA remains unclear. This study investigated the treatment efficiency of PS+RET on WS recovery among individuals with KOA and sarcopenia. A total of 108 older adults aged ≥ 60 years who had a diagnosis of radiographic KOA and sarcopenia were enrolled in this prospective cohort study. Sarcopenia was defined on the basis of the cutoff values of the appendicular skeletal muscle mass index for Asian people and a slow WS less than 1.0 m/s. The patients were equally distributed to three groups: PS+RET, RET alone, and usual care. The weekly assessment was performed during a 12-week intervention and a subsequent 36-week follow-up period. A cutoff of 1.0 m/s was used to identify successful recovery to normal WS. Kaplan–Meier analysis was performed to measure the survival time to normal WS among the study groups. Multivariate Cox proportional-hazards regression (CPHR) models were established to calculate the hazard ratios (HRs) of successful WS recovery and determine its potential moderators. After the 3-month intervention, PS+RET as well as RET obtained greater changes in WS by an adjusted mean difference of 0.18 m/s (*p* < 0.0001) and 0.08 (*p* < 0.05) m/s, respectively, compared to usual care. Kaplan–Meier analysis results showed both RET and PS+RET interventions yielded high probabilities of achieving normal WS over the 12-month follow-up period. Multivariate CPHR results revealed that PS+RET (adjusted HR = 5.48; *p* < 0.001), as well as RET (adjusted HR = 2.21; *p* < 0.05), independently exerted significant effects on WS recovery. PS+RET may accelerate normal WS recovery by approximately 3 months compared with RET. Sex and initial WS may influence the treatment efficiency. For patients with KOA who suffer sarcopenia, 12-week RET alone exerts significant effects on WS recovery, whereas additional PS further augments the treatment effects of RET by speeding up the recovery time of WS toward a level ≥ 1.0 m/s, which facilitates the patients to diminish the disease severity or even free from sarcopenia.

## 1. Introduction

Knee osteoarthritis (KOA), the most prevalent musculoskeletal disorder among older adults [1], is recognized as a major cause of chronic pain and disability [2,3,4,5]. KOA imposes a substantial burden on the physical health of affected individuals, resulting in significant economic costs to healthcare systems across regions and countries [6,7,8,9,10]. KOA is primarily characterized by pain and muscle weakness, which greatly affect the physical mobility, especially walking speed (WS), of the lower extremities [5,11,12]. Slow walking is considered an early marker of 5-year all-cause mortality rates [13] and is associated with high 3-year hospitalization rates [14]. In KOA, maximum WS is significantly correlated with quality of life [15]. Slow WS is associated with the long-term decline of physical activity [16], whereas relatively fast WS is associated with robust physical activity [17]. Therefore, maintenance or restoration of walking ability is crucial for individuals with KOA to sustain a high quality of life and avoid physical inactivity.

Sarcopenia, a condition characterized by age-related muscle attenuations in older adults, is clinically diagnosed by identifying low levels of muscle mass, strength, and physical performance [18,19,20]. Accordingly, an older individual who exhibits slow walking (i.e., low physical performance) may receive a diagnosis of probable, confirmed, or severe sarcopenia on the basis of the number of sex-specific muscle mass and strength criteria satisfied [18,19,20]. Elders with KOA characterized by impaired physical function, predominantly evident in a slow WS, are considered at a high risk of sarcopenia [21,22]. Observational studies have demonstrated that older individuals with KOA are more susceptible to sarcopenia than those without KOA [22,23,24,25]. In addition, the shared characteristics of low strength and slow WS strengthen the relationship between KOA and sarcopenia with respect to KOA incidence [26] and progression [23,24,27,28]. Further, the association between slow WS and reduced physical activity in KOA may lead to deconditioning [16,17,29,30], which in turn may increase the risk of gait decline [31] and accelerate the progressive loss of muscle mass, with an increased risk of sarcopenia [32]. Because a slow WS is one of the indicators of sarcopenia as well as disease severity [18,19,20], improving WS in older individuals with sarcopenia may mitigate disease severity and even reverse sarcopenia. Although sarcopenia is considered part of the normal aging process, its potential deleterious effects include limiting or reducing physical mobility; thus, successfully treating sarcopenia can provide an increasing number of older and very old adults with a higher quality of life. Therefore, establishing effective treatments for WS recovery in older individuals with KOA and sarcopenia is vital.

Because of the complex musculoskeletal cross-talk that occurs when KOA coexists with sarcopenia, developing a treatment regime targeting the comorbidities shared between KOA and sarcopenia is challenging [29,33,34]. Therefore, multidisciplinary approaches that incorporate pharmacological and nonpharmacological treatments have been recommended to treat the comorbidities associated with KOA. According to the evidence-based guidelines for the management of KOA [35,36,37], exercise therapy is strongly recommended for strength gain and function restoration and is deemed as a first-line treatment for KOA [38]. Among the various exercise training regimens for older individuals, resistance exercise training (RET) has demonstrated promising effects in terms of muscle mass gain, strength increase, and WS recovery in sarcopenia [39,40,41] and KOA [42,43]. In particular, an elastic RET using elastic strips or bands has been frequently used as a treatment method and is considered safe for muscle strengthening in elderly people with KOA [44,45,46]. In addition to exercise training, nutritional interventions such as protein supplementation (PS) are believed to augment the efficacy of exercise training for sarcopenia among older adults [47,48,49,50,51]. However, few studies have investigated the effects of PS plus RET (PS+RET) on older individuals with KOA [45]. WS may positively influence a detrimental effect of sarcopenia, namely dependency while performing activities of daily living [52], and therefore, increasing WS can increase the probability of recovery from disability [53] and reduce the need for long-term care [54]. However, additional evidence of the effect of PS+RET on the longitudinal trajectory of WS recovery in older adults with KOA, especially in those who receive a diagnosis of sarcopenia, is required.

The purpose of this study was to identify the effects of PS plus elastic RET on the time required to recover from slow walking, as well as its potential moderators, in individuals with KOA comorbid with sarcopenia. The study hypothesized that PS+RET would be more likely to result in successful WS recovery compared with RET alone.

## 2. Materials and Methods

### 2.1. Study Design

This prospective cohort study was conducted in accordance with the Strengthening the Reporting of Observation studies in Epidemiology (STROBE) guidelines [55]. The present study had three parallel study arms and enrolled patients from May 2019 to April 2022. The study protocol was approved by the Joint Institutional Review Board of Taipei Medical University (Trial number N201903004). All the enrolled patients provided written consent to participate at baseline admission before the intervention. The included patients were sequentially and equally allocated to three study groups: an intervention group receiving PS+RET, an intervention group receiving elastic RET alone, and an age-matched control group receiving usual care (UC). Data on the patients’ demographics and comorbidities were extracted through a standard medical chart review.

### 2.2. Participants

Older patients aged ≥60 years were recruited from the outpatient department of orthopedics and rehabilitation medicine at our hospital. All patients received a diagnosis of KOA according to the American College of Rheumatology criteria [56]. KOA severity was determined through radiographic assessment and was classified according to the Kellgren and Laurence (KL) grading system [57]. Knee pain was assessed using a 10-point visual analog scale. All patients who had a KL grade of 2–4 were included. After enrollment, data on the patients’ demographic characteristics and comorbidities were extracted through a standard medical chart review, and a comorbidity score for each patient was calculated using the Cumulative Illness Rating Scale (CIRS) [58]. The patients were excluded if they had any of the following conditions: (a) a history of hip or knee arthroplasty, (b) sensitivity or allergy to milk proteins or impaired renal function, (c) uncontrolled hypertension, (d) any cardiovascular or pulmonary disease that would prevent them from engaging in an exercise study, or (e) neurological or cognitive impairment. A flowchart of the patient selection process and study group assignment is illustrated in Figure 1.

### 2.3. Identification of Sarcopenia

Sarcopenia was identified on the basis of the criteria established by the Asian Working Group for Sarcopenia (AWGS) [19]. Low muscle mass was identified by an appendicular skeletal muscle mass (ASM) index of <7.0 kg/cm^2^ for Asian men and <5.7 kg/cm^2^ for Asian women. Bioelectrical impedance analysis (InBody 220, Biospace, Seoul, Republic of Korea) was performed to measure the ASM, which was further transformed to the ASM index by dividing the ASM value by the square of patient height. Slow WS was identified by a 6 m walk test pace of <1.0 m/s. Patients with low muscle mass and slow WS were classified as having sarcopenia.

Walking speed was measured using the 6 m walk test, following the 2019 recommendations of the AWGS [19]. All patients were asked to walk a distance of 6 m on a level surface at a comfortable speed. The time required to complete each trial of the walking task was recorded. Two trials were conducted for each assessment, and the mean value of the time measurements was calculated and converted to gait speed in m/s. Two research assistants who were blinded to the study group assignment assessed the WS at baseline and conducted weekly assessments of each patient during the study period. The reliability of the 6 m walk test has been determined to be high, as evidenced by an intraclass correlation coefficient of 0.92 for older adults [59].

### 2.4. Resistance Exercise Training

The exercise regime used in this study was based on our earlier established exercise protocol [60]. Elastic RET was employed using different colored elastic bands (Thera-Band products, Hygenic Co., Akron, OH, USA). The different colors of the elastic bands (i.e., yellow, red, green, blue, black, and silver) denote the degree of elasticity and the corresponding resistance level. All the participants were given an individualized exercise book containing instructions for and images of the exercise movements. Details on the exercise regime and exercise progression protocol are provided in Appendix A, respectively.

The elastic RET protocol comprised initial 2-week supervised training sessions (three sessions weekly) followed by 10 weeks of home-based exercise (one session daily). The initial 2 weeks of RET sessions were considered a familiarization period aimed at ensuring that each exercise movement would be performed correctly at home. During the supervised training sessions, all the exercise movements were supervised by a licensed senior physical therapist who was blinded to group assignments. The Borg Rating of Perceived Exertion scale was used to rate the patients’ perceived exertion during the training sessions [61,62]. Following the guidelines of the American College of Sports Medicine [63], an exercise intensity with a perceived effort ranging from moderate to high was the target for all participants in each exercise session (Appendix A). For each resistance grade, each participant was instructed to give an exertion level of “somewhat hard (moderate)” but not to exceed the level of “hard (high)”, with these levels corresponding to ratings of 13–15 on the Borg scale [61]. Each exercise session consisted of a 10 min warm up, movement of the upper and lower extremities, and a 10 min cool down (Appendix A). The exercise was progressively intensified by increasing the elasticity level of bands when the patients achieved their target Borg scale scores. We provided an exercise log book for each participant to record their progress over the exercise period. The physical therapist monitored patient compliance with the exercise regimen during the home exercise period every week through phone contact.

### 2.5. Assessment of Protein Intake and Dietary Protein Supplementation

Before the intervention started, each patient’s habitual daily protein intake was assessed using a 3-day food diary [64]. All participants were instructed to maintain their regular dietary habits and write in their food diaries. Food intake was recorded on 3 consecutive days, during which 1 day of the weekend was included. Dietary intake data were analyzed using the open-source software MyFitnessPal (MyFitnessPal, Inc., San Francisco, CA, USA) [65], which has been employed to analyze food intake data in previous studies [66,67].

The nutritional supplements (Affix Health Pte. Ltd., Singapore, Taiwan Branch) were in powder form and dissolved in water before consumption. Each serving contained 14 g of protein (7.0 g of whey, 4.2 g of casein, and 2.8 g of soy protein) and 7.3 g of amino acids (2.2 g of leucine). Individualized PS dose was determined on the basis of daily protein intake assessed at baseline to ensure the patients met the protein recommended dietary allowance of protein of 1.5 g/kg/day on the training days. The PS dose of 1.5 g/kg/day was chosen because it is an approximate amount for older adults who are undergoing RET, especially for those who have chronic diseases [68]. For example, a patient who weighed 60 kg and had a baseline daily protein intake of 48 g (i.e., 0.8 g/kg/day) was prescribed PS with an amount of 42 g (i.e., additional 0.7 g/kg/day) on each training day. The individualized PS doses were equally distributed in two servings (for the above example, 21 g per serving), and all the patients in the PS+RET group were instructed to consume two servings (one at breakfast and one within 1 h after RET) of protein powder on each training day during the intervention period. The participants were instructed to consume the same dose of supplements on the scheduled training days even if the RET was not performed due to any cause. A research assistant monitored participant compliance with the PS regimen every week through phone contact.

### 2.6. Usual Care

Patients in the UC group received conventional physical therapy that included electrical modalities, stretching exercises, and functional training not related to RET.

After 3 months of nutritional and exercise intervention, the patients in all 3 study groups were instructed to maintain their usual physical activity.

### 2.7. Identification of Successful Recovery from Slow Walking Speed

Recovery of walking ability was identified based on the WS, which is a determinant of the development of sarcopenia [19,20,69] and frailty [70,71]. Following the AWGS recommendations, a cutoff WS of 1.0 m/s was used to identify slow WS; accordingly, a WS of ≥1.0 m/s was defined as normal. All the patients enrolled in the present study demonstrated a WS of <1.0 m/s at baseline admission, indicating that they had low physical ability at the beginning of the study. For each patient, the time (in weeks) required to recover to normal WS (i.e., WS ≥ 1.0 m/s) was estimated on the basis of the weekly follow-up data.

### 2.8. Statistical Analysis

Continuous variables were assessed using a one-way analysis of variance to identify differences in the demographic characteristics between the study groups at baseline. Categorical variables were assessed using the chi-squared test. Post hoc analysis was performed using the Bonferroni correction, where equal variance was assumed. The Games–Howell test was performed if the homogeneity of group variance was not assumed. To identify any between-group differences in WS among 3 study groups after the 3-month intervention, an analysis of variance was performed using baseline WS as a covariate. All analyses were performed based on an intention-to-treat approach using the last observation carried forward method to impute any missing data [72].

Time-to-event analysis was performed to estimate the survival probability of successful recovery from slow WS. WS recovery was calculated using the Kaplan–Meier method. The log-rank (Mantel–Cox) test was employed to compare survival curves between the study groups.

To identify any moderators of treatment effects, a multivariate Cox proportional-hazards regression model was established to estimate hazard ratios (HRs) with corresponding 95% confidence intervals (CIs) by using age, sex, body mass index (BMI), disease severity (i.e., KL grade), comorbidity score (i.e., CIRS score), pain score, and WS at baseline as covariates. Based on the adjusted Cox regression model, the likelihood of achieving a successful recovery from slow walking was assessed using cumulative hazard estimates according to treatment groups and the identified potential moderators. All differences with *p* < 0.05 were considered statistically significant. SPSS Statistics (version 22.0; IBM, Armonk, NY, USA) was used for all analyses.

## 3. Results

### 3.1. Patient Demographics and Clinical Characteristics

The flow diagram of patient enrollment and allocation in this study is presented in Figure 1. After 1022 patients who did not fulfill the inclusion criteria and 13 who declined to participate were excluded, the sample finally comprised 108 patients. After obtaining written informed consent from all the included patients, the patients were equally allocated to the PS+RET (*n* = 36), RET (*n* = 36), and UC (*n* = 36) groups at baseline. Over the study period, three patients withdrew from the study at the 3-month intervention, and three were lost to follow-up after the intervention. No dropout event was related to the interventions. The patient demographics and characteristics are presented in Table 1. Overall, the mean (range) age, mean (range) BMI, and mean (range) disease duration of the study sample was 73.9 (61–86) years, 27.9 (19.8–37.2) kg/m^2^, and 9.9 (0.6–37.8) years, respectively. In addition, the mean daily habitual protein intake was 0.81 (range 0.34–1.28) g/kg/day among all of the included patients. The characteristics exhibited no significant differences between the three groups (all *p* > 0.05).

### 3.2. Intervention Effects on Walking Speed

After the 3-month intervention, patients in the PS+RET group walked by an adjusted mean WS of 0.96 (95% CI: 0.91–1.02) m/s faster than their peers who received UC did (adjusted mean WS = 0.78 (95% CI: 0.74–0.84) m/s; *p* < 0.001), with a corresponding mean difference of 0.18 (95% CI: 0.10–0.25) m/s. Similar results were observed in the RET group; those who received RET alone obtained greater changes in WS and responded to 3-month exercise training with an adjusted mean difference of 0.08 (95% CI: 0.01–0.15) m/s, compared to the UC group.

### 3.3. Survival Time for Recovery from Slow Walking Speed

Overall, 55 (50.9%) patients in the study cohort, specifically 25 (69.4%) from the PS+RET group, 20 (55.6%) from the RET group, and 10 (27.8%) from the UC group, achieved a WS of ≥1.0 m/s over the 12-month study duration.

The results of the Kaplan–Meier analysis revealed an overall median time of 26 weeks for all study patients to achieve a WS of ≥1.0 m/s (Figure 2A). Results of the log-rank (Mantel–Cox) test indicated a significant difference between the survival curves at 52 weeks (log-rank *p* < 0.0001; Figure 2B). Over the 12-month study period, patients in the PS+RET (log-rank *p* < 0.0001) and RET (log-rank *p* = 0.02) groups had a higher probability of successful WS recovery compared with the control group. In addition, the patients who received PS+RET had a shorter time (median survival time = 14 (95% CI, 10.5–17.5) weeks) to recovery to normal WS compared with their counterparts in the RET (median survival time = 25 (95% CI, 16.9–33.1) weeks; log-rank *p* = 0.04) group.

### 3.4. Effects of Interventions on Time to Successful Walking Speed Recovery and Its Moderators

Table 2 presents the results of the Cox regression analyses demonstrating the effects of RET and PS+RET on the time to recovery of normal WS as well as its potential moderators. Figure 3 presents the cumulative hazard estimates of the likelihood of successful WS recovery stratified by the treatment group and the identified potential moderators.

The crude model indicated that PS+RET yielded an HR of 4.21 for time to recovery from slow WS, which was nearly twofold higher than that in the RET group (crude HR = 2.27) compared with the UC control. After adjustment for age, sex, BMI, disease severity, comorbidity score, pain score, and WS at baseline (Table 2), the rate of recovery of WS in the PS+RET group increased, with an adjusted HR of 5.48; similar results were observed for the RET group (adjusted HR = 2.54). Additionally, sex (adjusted HR = 2.12) and WS at baseline (adjusted HR = 21.63) were significant moderators influencing the treatment effects on the WS recovery rate. Male patients (adjusted HR = 2.12) who walked faster at baseline (adjusted HR = 21.63) were observed to spend less time achieving normal WS compared with their counterparts who walked slower at baseline measurement (Table 2).

The results of the cumulative hazard analysis based on the multivariate Cox regression model indicated an increased likelihood of achieving a successful recovery from slow WS beyond 12 months for patients in the PS+RET and RET groups compared with those in the UC group (Figure 3A) as well as for male patients compared with their female peers (Figure 3B).

### 3.5. Compliance with Protein Supplementation and Exercise

All participants in the PS+RET group were confirmed to have completed their supplementation regimens, corresponding to a 100% adherence rate. No participant discontinued supplementation because of discomfort or gastrointestinal complaints.

On average, compliance with the exercise program in the RET and PS+RET groups was 79.8% and 81.2% of the expected exercise sessions, respectively. No serious side effects were reported during the exercise intervention period. During three months of exercise intervention, a total of five patients (two in PS+RET; three in RET) reported knee pain after exercise training. All of the nonserious events were improved by ice packing and resting, and thereafter, none needed pain medications. Accordingly, the intensity of RET for the five patients was adjusted to an exertion level of “light” corresponding to ratings of 10–12 on the 15-point Borg scale [61] in their progress of RET protocol.

## 4. Discussion

### 4.1. Summary of Main Findings

This study determined the effects of PS+RET and its moderators on the time required to recover from slow walking in older individuals with KOA comorbid with sarcopenia. The results indicated the following: (1) PS+RET, as well as RET alone, exerted significant effects on changes in WS after the 12-week intervention; (2) PS+RET, as well as RET alone, resulted in a higher probability of successful WS recovery compared with UC; (3) the patients receiving PS+RET experienced a shorter time to achieve a normal WS (i.e., ≥1.0 m/s) than those receiving RET alone; (4) after adjustment for potential moderators, namely age, sex, BMI, disease severity, comorbidity score, pain score, and WS at baseline, the patients exhibited higher cumulated hazards of successful WS recovery in response to PS+RET than in response to RET and UC; and (5) sex and initial WS may contribute to the efficacy of treatment for WS recovery.

### 4.2. Sarcopenia Prevalence in Study Cohort

Population-based cohort studies have demonstrated an overall prevalence of primary sarcopenia ranging from 8.3% to 40.0% among individuals with KOA [23,24,32,73,74,75,76]. Such a high prevalence of sarcopenia may be due to the different criteria used in these studies for identifying sarcopenia. According to the 2019 AWGS updated consensus on diagnostic criteria for sarcopenia, 121 of the 1143 patients enrolled in this study were identified as having sarcopenia, for a prevalence of 10.6%. This finding is consistent with those of previous studies that reported a sarcopenia prevalence of 9.1–11.8% in patients with KOA according to composite criteria of low muscle mass combined with low WS [23,32,74]. When the single criterion of low muscle mass was considered, the prevalence of sarcopenia reached 40.0% [73,76].

### 4.3. WS Recovery Pattern over the Follow-Up Period

In this study, patients who achieved normal WS (i.e., ≥1.0 m/s) over the 12-month follow-up period were considered to have achieved successful WS recovery. In the RET group (Figure 2B), 5 out of 36 (13.9%) patients achieved normal WS during the 12-week training period, and an additional 15 (41.7%) patients achieved normal WS over the 16-week detraining period (i.e., from the 12th to 28th week). Thereafter (i.e., from the 28th to the 52nd week), no patient achieved normal WS. Among the patients who received PS+RET, 14 (38.9%) exhibited successful WS recovery until the end of the intervention, and an additional 11 (30.6%) patients experienced normal WS over the 6-week detraining period (i.e., from the 12th to 18th week); thereafter (i.e., from the 18th to 52nd week), no patient achieved normal WS. Our findings indicate that additional PS may accelerate WS recovery by 11 weeks for patients with KOA who undergo RET and that WS recovery in response to interventions is time-dependent. Therefore, PS+RET may be an efficient treatment strategy, particularly for achieving WS recovery in older patients with KOA and sarcopenia.

In regard to the WS recovery pattern, our results revealed that the trajectory of WS recovery reached a peak during 6–16 weeks after the intervention, indicating a short- to moderate-term effect. Grgic [77] conducted a systematic review to investigate the effectiveness of RET in older people and found that muscle size increased during RET periods, was maintained 12–24 weeks after RET, and decreased 31–52 weeks after RET. Considering that treatment-induced changes in muscle mass may contribute to the recovery of walking ability among populations with sarcopenia and frailty [78], Grgic’s results may support our findings that a majority of instances of successful WS recovery occurred within 16 weeks after treatment and that none were observed from the 28th to 52nd week.

### 4.4. Effect of RET on Time to Normal WS

Previous systematic review and meta-analysis studies have revealed the efficacy of RET in helping older adults with sarcopenia successfully recover WS, with an overall effect size of 1.50–2.01 on WS recovery [41,79]. In the present study, on older patients with KOA who had sarcopenia, the RET group had a higher probability of WS recovery (adjusted HR = 2.21) compared with the UC group, with an estimated effect size of 0.62 on time to recover a normal WS [80,81]. The small effect size of speeding WS recovery time in this study may indicate that the presence of KOA as a comorbidity has an impact on the effect of treatment with RET in people with sarcopenia. Because knee pain significantly affects gait pattern, which results in a decline in walking performance [82], and is a predictor of poor mobility in older adults [83], sarcopenic older adults with KOA may experience negligible WS recovery in response to RET compared with their peers without KOA.

A systematic review by Li et al. [43] indicated that RET exerted favorable effects on WS by an effect size of 0.03 for WS restoration among older people with KOA and without sarcopenia; this effect size of WS restoration was smaller than that observed in our patients with KOA and sarcopenia. These findings indicate that patients with both sarcopenia and KOA achieve greater WS outcomes in response to RET than those without sarcopenia do, although minor changes in WS were reported after postoperative rehabilitation in patients with KOA and sarcopenia who had undergone total knee replacement [84]. Regarding the measurement of WS outcomes, all the trials included in Li’s systematic review compared the training-induced changes in WS between RET and control groups, whereas the present study statistically tested the difference in time to recovery to normal WS between the RET and UC groups. Therefore, the treatment efficiency of RET appears clinically relevant rather than its efficacy for older individuals with KOA and sarcopenia, especially in terms of WS recovery, which is time-dependent.

### 4.5. Treatment Efficiency of PS+RET on Time to Recovery to Normal WS

To facilitate the translation of evidence-based recommendations into clinical practice for sarcopenia and KOA, several exercise and nutrition regimens have been developed for the delivery of disease care. Among the various exercise modes and nutritional supplements, additional PS is believed to augment the treatment effects of RET among older people with sarcopenia and frailty. As previously reviewed [47,48,49,51,78,85,86,87], the PS+RET intervention provides obvious benefits to older adults with sarcopenia. However, the clinical outcome in terms of the trajectory of WS recovery over a long-term follow-up period remains unclear. In addition, PS+RET has been less frequently applied to those with KOA, and thus, treatment compliance responding to PS+RET in those who concurrently have KOA and sarcopenia remains unclear. Therefore, in this study, we recruited older patients with KOA who had sarcopenia and demonstrated that PS+RET was well tolerated by this population, with significant effects on the WS recovery rate.

In the present study, the PS+RET intervention yielded successful WS recovery, with an adjusted HR of 5.48, approximately 2.5 times higher than that of the RET group. Accordingly, the effect size of PS+RET for WS recovery can be estimated as 0.48 compared with that of RET and 1.32 compared with that of UC [80,81]. Our results are supported by previous systematic reviews, which reported that PS+RET exerts favorable effects on WS recovery, with a pooled effect size of 0.10–0.25 [48,78,85] and 0.41–4.14 [78,86,87] for restoration WS, compared with RET and nonexercise controls, respectively.

Notably, according to the analysis results in this study, recovery to a WS of ≥1.0 m/s was faster (median survival time = 14 weeks) in those treated with PS+RET than in those performing RET alone (median survival time = 25 weeks). These results indicate that the incorporation of additional PS into the RET regime may shorten the time to recovery to normal WS by approximately 3 months. One possible reason for this is that incorporating PS into the RET intervention can enhance the effects of RET on the increase in ASM among older people with high sarcopenia risk [51,78,88,89], and, in turn, ASM may contribute to the regain of leg strength and walking capability during RET [78]. Thus, PS+RET is more likely to accelerate WS recovery than RET alone. Based on the recommendations of the European Society for Clinical Nutrition and Metabolism Expert Group [90], our findings support the urgent need to combine PS with RET to prevent functional decline in older adults with KOA and sarcopenia.

### 4.6. Moderators Influencing Treatment Efficiency

In the present study, male patients were 2.1 times more likely to achieve normal WS than female patients. Although one study reported sex differences in the adaptations of older adults with KOA to RET in terms of muscle mass and strength gains [91], others have not [92,93,94]. Such sex-specific muscle morphological and functional adaptations to RET are more apparent in the lower extremities [92,93], which further contributes to physical performance. Therefore, the results described above may explain our finding of a sex-specific WS recovery pattern regardless of the intervention.

Alternatively, several systemic reviews have indicated that sex is a nonsignificant moderator of treatment efficacy on muscle mass, strength, and physical performance for PS+RET and RET regimens [49,78]. These results indicate that intergroup differences (i.e., PS+RET versus RET) in treatment-induced muscular adaptations are independent of sex. Because of our limited sample size, we could not conduct a sex-stratified analysis to assess whether sex is a confounding factor. However, the results of previous studies support our findings that the treatment effectiveness of both PS+RET and RET is independent of sex. Future studies conducted with a sample size larger than 180 participants, among which two sexes are equally (i.e., 30 men and 30 women) assigned to each study group, are warranted to identify whether sex has any effect on the treatment efficacy of additional PS in older individuals who are undergoing RET.

Another notable finding was that the baseline WS appeared to moderate the recovery rate over the long-term follow-up in this study. This finding indicates that patients who had a faster WS at baseline required less time to achieve normal WS. This result accords with a study that concluded that older adults with KOA who walk faster are 1.9 to 2.0 times more likely to achieve a minimal clinically important improvement in physical function compared with their peers who are slower over a 30-month follow-up period [95].

### 4.7. Study Limitations

Several limitations of this study must be considered. First, this was a prospective cohort study, and the allocation of study groups was not randomized. Therefore, potential selection bias must be considered when interpreting the results. Further randomized controlled studies are warranted to minimize this selection bias. Second, placebo supplementation was not employed in the RET or UC groups. Therefore, blinding the patients to group assignment was difficult, which may have resulted in performance bias. Third, lifestyle behaviors, such as physical activity level, may have contributed to long-term WS outcomes [16,31]. Physical activity was not assessed or included as a covariate in the analysis; therefore, differences in the probability of successful WS recovery between the treatment groups may have been overestimated. Finally, the use of pharmacological medications and intra-articular injections for pain control was allowed during the intervention period. Because such potential confounders of long-term WS recovery were not considered, the time to recover to normal WS after treatment may be overestimated. Future studies should evaluate whether pain medications or intra-articular injections have a significant intragroup effect on the WS recovery rate after treatment.

## 5. Conclusions

This study demonstrated that PS+RET and RET alone were associated with significantly obvious changes in WS after a 12-week intervention and high probabilities of achieving normal WS over a 12-month follow-up period among patients with KOA who had sarcopenia. The study results further indicated that additional PS employed during the 12-week intervention of RET may reduce the time required to achieve normal WS by approximately 3 months. Finally, analyses of potential moderators revealed that the male sex and a higher WS at baseline might predict a higher probability of successful recovery to normal WS. The results of the current study have implications for clinicians, namely that rehabilitation strategies should be devised to optimize WS restoration in patients with KOA, especially those who also have sarcopenia.

## Figures and Tables

**Figure 1 nutrients-15-01552-f001:**
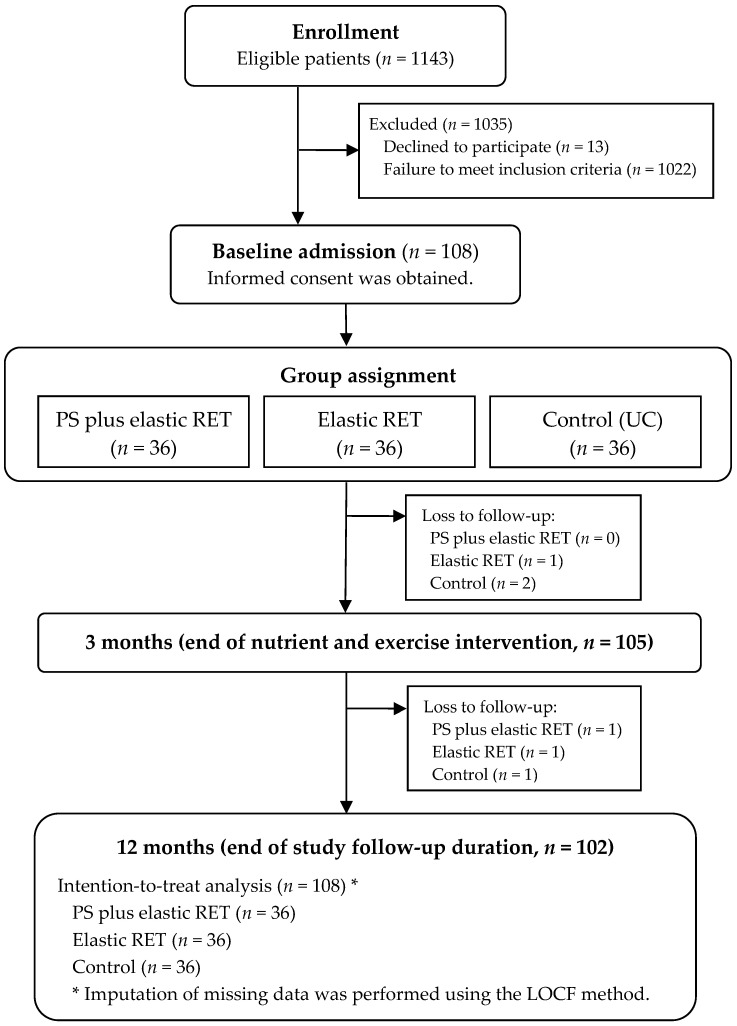
Flowchart of patient enrollment and allocation in the present study. PS, protein supplementation; RET, resistance exercise training; UC, usual care; LOCF, last observation carried forward.

**Figure 2 nutrients-15-01552-f002:**
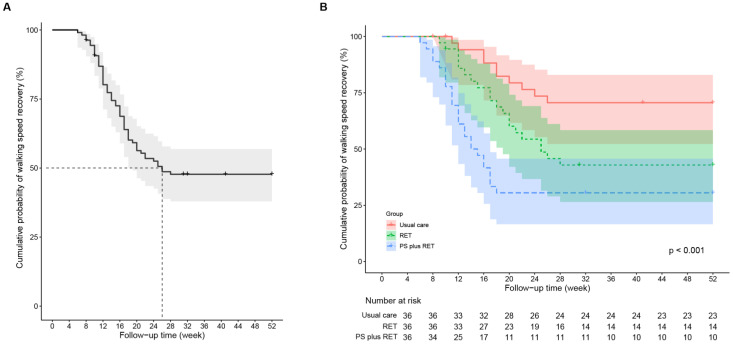
Kaplan–Meier survival curve for time to recovery to normal walking speed (**A**) for the overall study cohort and (**B**) by treatment group. PS, protein supplementation; RET, resistance exercise training.

**Figure 3 nutrients-15-01552-f003:**
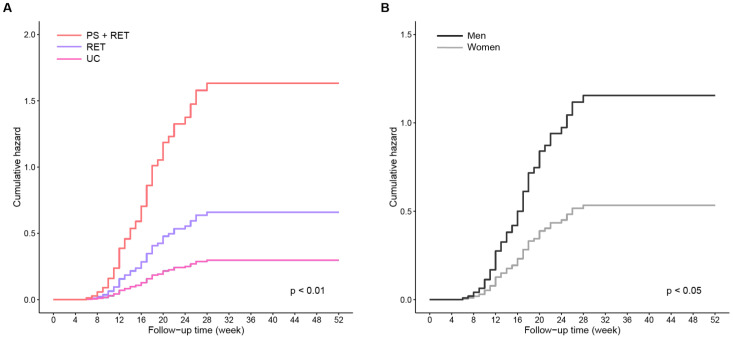
Cumulative hazard estimates of the likelihood of achieving normal walking speed according to (**A**) treatment group and (**B**) sex. All estimates are based on the results of the multivariate Cox proportional-hazards model with age, disease severity, comorbidity score, pain score, and walking speed at baseline used as covariates.

**Table 1 nutrients-15-01552-t001:** Baseline characteristics of study participants.

Characteristic	UC	RET	PS+RET	*p*
	Mean ± SD	Mean ± SD	Mean ± SD	
*n* ^a^	36	36	36	
Women, *n* (%)	23(63.9)	26(72.2)	25(69.4)	0.740 ^c^
Age (years)	74.9 ± 6.6	73.8 ± 5.7	73.5 ± 6.3	0.540 ^b^
ASMMI (kg/m^2^)				
Men	6.2 ± 0.6	6.3 ± 0.6	6.2 ± 0.7	0.973 ^b^
Women	5.1 ± 0.4	5.1 ± 0.5	5.1 ± 0.5	0.996 ^b^
BMI (kg/m^2^)	27.4 ± 3.3	28.2 ± 3.4	28.1 ± 3.7	0.567 ^b^
CIRS score	14.9 ± 4.6	15.2 ± 4.7	13.7 ± 4.4	0.376 ^b^
Disease duration (year)	10.8 ± 7.9	9.2 ± 8.2	9.9± 8.6	0.714 ^b^
Involved leg, *n* (%)				0.856 ^c^
Right	20(55.6)	18(50.0)	22(61.1)	
Left	9(25.0)	12(33.3)	9(25.0)	
Bilateral	7(19.4)	6(16.7)	5(13.9)	
KL grade, *n* (%)				0.612 ^c^
2	7(19.4)	11(30.6)	10(27.8)	
3	24(66.7)	19(52.8)	18(50.0)	
4	5(13.9)	6(16.7)	8(22.2)	
Number of comorbidities, *n* (%)				0.789 ^c^
1	11(30.6)	11(30.6)	12(33.3)	
2	11(30.6)	11(30.6)	14(38.9)	
3	8(22.2)	11(30.6)	6(16.7)	
>3	6(16.7)	3(8.2)	4(11.1)	
Protein intake (g/kg/day)	0.84 ± 0.26	0.79 ± 0.25	0.81 ± 0.24	0.699 ^b^
Pain (VAS)	7.5 ± 1.4	7.1 ± 1.4	7.7 ± 1.4	0.193 ^b^
Walking speed (m/s)	0.72 ± 0.16	0.75 ± 0.14	0.73 ± 0.15	0.743 ^b^

^a^ All analyses were based on an intention-to-treat analysis. ^b^ One-way analysis of variance. ^c^ Chi-squared test. ASMMI, appendicular skeletal muscle mass index; BMI, body mass index; CIRS, Cumulative Illness Rating Scale; KL grade, Kellgren–Lawrence grade; PS, protein supplementation; RET, resistance exercise training; SD, standard deviation; UC, usual care; VAS, visual analog scale.

**Table 2 nutrients-15-01552-t002:** Results of Cox proportional-hazards model imputing potential moderators for time to successful walking speed recovery at 12-month follow-up.

Model Established	Coefficient (B)	SE (B)	Hazard Ratio	95% CI (Lower, Upper)	*p*
Covariates					
Crude model (*n* = 108; overall χ^2^ = 16.91, *p* < 0.001)					
Group (ref: UC)					
RET	0.82	0.39	2.27	(1.06, 4.86)	0.03
PS plus RET	1.44	0.38	4.21	(2.01, 8.79)	<0.001
Adjusted model (*n* = 108; overall χ^2^ = 44.75, *p* < 0.001)					
Group (ref: UC)					
RET	0.79	0.40	2.21	(1.01, 4.82)	0.04
PS plus RET	1.67	0.39	5.48	(2.48, 11.23)	<0.001
Sex (ref: women)					
Men	0.75	0.32	2.12	(1.13, 3.97)	0.02
KL grade (ref: grade 2)					
Grade 3	−0.14	0.33	0.87	(0.46, 1.65)	0.67
Grade 4	−0.26	0.49	0.77	(0.29, 2.03)	0.60
Age	–0.05	0.03	0.95	(0.90, 1.01)	0.11
BMI	–0.04	0.04	0.96	(0.88, 1.05)	0.39
CIRS score	–0.02	0.03	0.98	(0.92, 1.05)	0.61
Pain score (baseline)	–0.07	0.11	0.93	(0.75, 1.16)	0.51
Walking speed (baseline)	3.07	1.22	21.63	(1.97, 237.09)	<0.01

SE, standard error; 95% CI, 95% confidence interval; UC, usual care; RET, resistance exercise training; PS, protein supplementation; KL grade, Kellgren–Lawrence grade; BMI, body mass index; CIRS, Cumulative Illness Rating Scale.

## Data Availability

The data presented in this study are available on request from the corresponding author. Public data sharing is not applicable to this article due to ethical considerations and privacy restrictions.

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
