# Peer review of "Effects of Protein Supplementation Combined with Resistance Exercise Training on Walking Speed Recovery in Older Adults with Knee Osteoarthritis and Sarcopenia"

_nutrients, 2023, doi:10.3390/nu15071552_

Round 1

Reviewer 1 Report

The authors may want to mention that STROBE guidelines were used for reporting of the study.

In the title, I suggest to change “exercise” to “exercise training”.

L30. “slow WS”. I suggest to provide a value.

In the abstract, I suggest to provide quantitative information on the walking speeds.

The conclusion of the abstract seems to repeat the observations of the study. I suggest to provide a conclusion on the application of your findings for the cohort of interest.

In the introduction, there is no mention of the type of resistance exercise training. I suggest to clarify in the introduction that elastic resistance exercise training is the training modality.

L108. Was the consent written or oral? Please clarify.

In Figure 1, from 3 to 12 months there is a loss of 3 but number changes from 105 to 101. Please check.

L182. Was the protein supplementation strategy based on previous work. In addition, it was not individualized, was that on purpose?

Figure 1 indicates n=108 at 12-months but Table 1 indicates n=108. Please clarify/revise.

L291. “no serious side effects”. So, that assumes there were side effects reported. The authors may want to provide those and how many individuals did not report any side effects.

L313. Change “67] When” to “67]. When”

L344 and throughout the manuscript, clarify which effect size is presented.

L355. Please present the effect size you are referring to, as no effect sizes are presented.

L414. The authors may want to suggest what the sample size of future studies should be.

Author Response

Reviewer 1

Comments and Suggestions for Authors

Response

We thank all the reviewers for their comprehensive review and their comments regarding our manuscript. We have made all necessary modifications to our originally submitted manuscript (Manuscript ID: nutrients-2290020), based on reviewers’ comments, point by point.

The authors may want to mention that STROBE guidelines were used for reporting of the study.

Response

Following the reviewer’s comment, a statement was added to mention that we followed the STROBE guidelines for reporting of this study.

Section 2.1 (Lines 110–111).

“This prospective cohort study was conducted in accordance with the Strengthening the Reporting of Observation studies in Epidemiology (STROBE) guidelines [56].”

In the title, I suggest to change “exercise” to “exercise training”.

Response

Following the comment, we revised the title as follows:

“Effects of protein supplementation combined with resistance exercise training on walking speed recovery in older adults with knee osteoarthritis and sarcopenia”

L30. “slow WS”. I suggest to provide a value.

Response

We revised the statement as follows:

Line 30

“…… and a slow WS less than 1.0 m/s.”

In the abstract, I suggest to provide quantitative information on the walking speeds.

Response

Based on the constructive comment, we added quantitative information on the walking speed (WS). We performed additional analyses to identify between-group difference in WS after 3-month intervention and added a section 3.2 in Results to describe the related results. The revised statements were summarized as follows:

Abstract

Lines 36–38

“After 3-month intervention, PS+RET as well as RET obtained greater changes in WS by an adjusted mean difference of 0.18 m/s (p < 0.0001) and 0.08 (p < 0.05) m/s, respectively, compared to usual care.”

Materials and Methods

Section 2.8 (Lines 232–234).

“To identify any between-group difference in WS among three study groups after 3-month intervention, analysis of variance was performed using baseline WS as a covariate.”

Results

Section 3.2 (Lines 267–273).

“After 3-month intervention, patients in the PS+RET group walked by an adjusted mean WS of 0.96 (95% CI: 0.91–1.02) m/s faster than their peers who received UC did [adjusted mean WS = 0.78 (95% CI: 0.74–0.84) m/s; P < 0.001], with a corresponded mean difference of 0.18 (95% CI: 0.10–0.25) m/s. Similar results were observed in the RET group; those who receiving RET alone obtained greater changes in WS responded to 3-month exercise training with an adjusted mean difference of 0.08 (95% CI: 0.01–0.15) m/s, compared to the UC group.”

Discussion

Section 4.1 (Lines 336–337).

“The results indicated the following: (1) PS+RET as well as RET alone exerted significant effects on changes in WS after 12-week intervention.”

Conclusions

Lines 485–488.

“This study demonstrated that PS+RET as well as RET alone were associated with significantly obvious changes in WS after 12-week intervention and high probabilities of achieving normal WS over a 12-month follow-up period among patients with KOA who had sarcopenia.”

The conclusion of the abstract seems to repeat the observations of the study. I suggest to provide a conclusion on the application of your findings for the cohort of interest.

Response

We corrected the statement as follows:

Lines 43–46.

“For patients with KOA who suffer sarcopenia, 12-week RET alone exerts significant effects on WS recovery whereas additional PS further augments the treatment effects of RET by speeding up the recovery time of WS toward a level ≥ 1.0 m/s, which facilitates the patients to diminish the disease severity or even free from sarcopenia.”

In the introduction, there is no mention of the type of resistance exercise training. I suggest to clarify in the introduction that elastic resistance exercise training is the training modality.

Response

Following the reviewer’s comment, we added statements to mention the exercise-training type as follows:

Lines 92–94.

“Especially, an elastic RET using elastic strips or bands has been frequently used as a treatment method and is considered safe for muscle strengthening in elderly people with KOA [44-46].”

Lines 104–105.

“The purpose of this study was to identify the effects of PS plus elastic RET on the time required to recover from slow walking, ……”

L108. Was the consent written or oral? Please clarify.

Response

We revised the statement as follows:

Section 2.1 (Lines 114–115).

“All the enrolled patients provided the written consent to participate at baseline admission before the intervention.”

In Figure 1, from 3 to 12 months there is a loss of 3 but number changes from 105 to 101. Please check.

Response

We corrected the number of patients at 12-month follow up as 102 in Figure 1 (Page 4).

L182. Was the protein supplementation strategy based on previous work. In addition, it was not individualized, was that on purpose?

Response

Thanks for the reviewer’s insightful comment, we added statements to clarify the individualized protein supplementation strategy as follows:

Section 2.5 (Lines 197–205).

“Individualized PS dose was determined on the basis of daily protein intake assessed at baseline to ensure the patients meting the protein recommended dietary allowance of protein of 1.5 g/kg/day on the training days. The PS dose of 1.5 g/kg/day was chosen because it is an approximate amount for older adults who are undergoing RET, especially for those who have chronic diseases [69]. For example, a patient who weighted 60 kg and had baseline daily protein intake of 48g (i.e., 0.8 g/kg/day) was prescribed PS with an amount of 42 g (i.e., additional 0.7 g/kg/day) on each training day. The individualized PS doses were equally distributed in two servings (for the above example, 21 g per serving) and ……”

Figure 1 indicates n=108 at 12-months but Table 1 indicates n=108. Please clarify/revise.

Response

Thank you. We performed all analyses based on an intention-to-treat approach. Therefore, all included patients at baseline (n = 108) were analyzed throughout the manuscript. We revised Figure 1 (Page 4) and revised statements to clarify the intention-to-treat approach as follows:

Section 2.8 (Lines 232–235).

“All analyses were performed based on an intention-to-treat approach using the last observation carried forward method to impute any missing data [73].”

L291. “no serious side effects”. So, that assumes there were side effects reported. The authors may want to provide those and how many individuals did not report any side effects.

Response

We provided information regarding nonserious side effects as follows:

Section 3.5 (Lines 326–331).

“During 3 months of exercise intervention, a total of five patients (two in PS+RET; three in RET) reported knee pain after exercise training. All of the nonserious events were improved by ice packing and resting, and thereafter none needed pain medications. Accordingly, the intensity of RET for the five patients were adjusted to an exertion level of “light”, corresponding to ratings of 10–12 on the 15-point Borg scale [62], in their progress of RET protocol.”

L313. Change “67] When” to “67]. When”

Response

We corrected the statement as follows:

Section 4.2 (Lines 353–354).

“according to a composite criteria of low muscle mass combined with low WS [23, 32, 75].”

L344 and throughout the manuscript, clarify which effect size is presented.

L355. Please present the effect size you are referring to, as no effect sizes are presented.

Response

We revised the statements to clarify which effect size is presented throughout the manuscript. The revised statements were listed as follows:

Section 4.4 (Lines 382–389).

“Previous systematic review and meta-analysis studies have revealed the efficacy of RET in helping older adults with sarcopenia successfully recover WS, with an overall effect size of 1.50–2.01 on WS recovery [41, 80]. In the present study on older patients with KOA who had sarcopenia, the RET group had a higher probability of WS recovery (adjusted HR = 2.21) compared with the UC group, with an estimated effect size of 0.62 on time to recover a normal WS [81, 82]. The small effect size of speeding WS recovery time in this study may indicate that the presence of KOA as a comorbidity has an impact on the effect of treatment with RET in people with sarcopenia.”

Section 4.4 (Lines 394–397).

“A systematic review by Li et al. [43] indicated that RET exerted favorable effects on WS by an effect size of 0.03 for WS restoration among older people with KOA and without sarcopenia; this effect size of WS restoration was smaller than that observed in our patients with KOA and sarcopenia.”

Section 4.5 (Lines 423–427).

“Accordingly, the effect size of PS+RET for WS recovery can be estimated as 0.48 compared with that of RET and 1.32 compared with that of UC [81, 82]. Our results are supported by previous systematic reviews, which reported that PS+RET exerts favorable effects on WS recovery, with a pooled effect size of 0.10–0.25 [48, 79, 86] and 0.41–4.14 [79, 87, 88] for restoration WS, ……”

L414. The authors may want to suggest what the sample size of future studies should be.

Response

We provided information to suggest a larger sample size of future as follows:

Section 4.6 (Lines 456–459).

“Future studies conducted with a sample size larger than 180 participants, among which two sexes are equally (i.e., 30 men and 30 women) allocated into study groups, are warranted to identify whether sex has any effect on the treatment efficacy of additional PS among older individuals undergoing RET.”

Reviewer 2 Report

In my opinion, the manuscript is prepared correctly. However, I have an objection to the methodology, which in my opinion is a serious limitation of the study.

 The comments on specific sections of the manuscript:

1) The introductory section explains the study design. The authors justify the research topic well.

2) The methodological chapter is correctly described. However, I would like the authors to explain why protein intake was not assessed in the study group. As is well known, this macronutrient is extremely important in the treatment of sarcopenia. Therefore, it seems reasonable to know the protein intake from foods when recommending the use of dietary supplements.

3) The descriptions of the results were correct, consistent with the common description of manuscripts narrative review

4) The conclusions were well formulated. I agree with this conclusions reached by the Authors.

Author Response

Reviewer 2

Comments and Suggestions for Authors

In my opinion, the manuscript is prepared correctly. However, I have an objection to the methodology, which in my opinion is a serious limitation of the study.

Response

We thank all the reviewers for their comprehensive review and their comments regarding our manuscript. We have made all necessary modifications to our originally submitted manuscript (Manuscript ID: nutrients-2290020), based on reviewers’ comments, point by point.

The comments on specific sections of the manuscript:

1) The introductory section explains the study design. The authors justify the research topic well.

2) The methodological chapter is correctly described. However, I would like the authors to explain why protein intake was not assessed in the study group. As is well known, this macronutrient is extremely important in the treatment of sarcopenia. Therefore, it seems reasonable to know the protein intake from foods when recommending the use of dietary supplements.

Response

Based on the constructive comment, we added information regarding daily habitual protein intake in the study groups. Accordingly, we made statements to clarify the assessment of daily habitual protein intake as follows:

Section 2.5 (Lines 187–193).

“Before intervention started, each patient’s habitual daily protein intake was assessed using a 3-day food diary [65]. All participants were instructed to maintain their regular dietary habits and write in their food diary. Food intake was recorded on 3 consecutive days during which 1 day of weekend was included. Dietary intake data were analyzed using the open-source software MyFitnessPal (MyFitnessPal, Inc., San Francisco, CA, USA) [66], which has been employed to analyze food intake data in previous studies [67, 68].”

In addition, the related results were added in Results section 3.1 and Table 1 (Page 7) as follows:

Section 3.1 (Lines 261–262).

“In addition, the mean daily habitual protein intake was 0.81 (range 0.34-1.28) g/kg/day among all of the included patients.”

3) The descriptions of the results were correct, consistent with the common description of manuscripts narrative review

4) The conclusions were well formulated. I agree with this conclusions reached by the Authors.
